# Role of Ceramide Kinase/C1P in the Regulation of Cell Growth and Survival

**DOI:** 10.3390/ijms26178374

**Published:** 2025-08-28

**Authors:** Ana Gomez-Larrauri, Asier Benito-Vicente, Asier Larrea-Sebal, César Martín, Antonio Gomez-Muñoz

**Affiliations:** 1Department of Biochemistry and Molecular Biology, Faculty of Science and Technology, University of the Basque Country (UPV/EHU), 48980 Bilbao, Bizkaia, Spain; ana.gomezlarrauri@osakidetza.eus (A.G.-L.); asier.benito@ehu.eus (A.B.-V.); asier.larrea@ehu.eus (A.L.-S.); cesar.martin@ehu.eus (C.M.); 2Respiratory Department, Cruces University Hospital, 48903 Barakaldo, Bizkaia, Spain; 3Department of Molecular Biophysics, Biofisika Institute, University of the Basque Country (UPV/EHU), Consejo Superior de Investigaciones Científicas (CSIC), 48940 Leioa, Bizkaia, Spain

**Keywords:** apoptosis, cancer, ceramide kinase, ceramide 1-phosphate, cell proliferation, cell survival, inflammation

## Abstract

Ceramide 1-phosphate (C1P) is a key regulator of cell proliferation and survival in both normal and transformed cells. Major pathways implicated in the mitogenic actions of C1P include activation of the mitogen-activated protein kinases (MAPKs) ERK1-2 and JNK, as well as stimulation of the phosphatidylinositol 3 kinase (PI3K)/Akt/mammalian target of rapamycin (mTOR) pathway, the product of retinoblastoma, or the sphingomyelin synthase (SMS)/diacylglycerol (DAG)/protein kinase C-alpha (PKC-α) pathway. C1P-stimulated cell proliferation can also be mediated through enhanced secretion of vascular endothelial growth factor (VEGF) in macrophages or by releasing lysophosphatidic acid (LPA) in myoblasts. Also, the production of low levels of reactive oxygen species (ROS) can mediate the stimulation of cell growth by C1P, particularly in macrophages. Upregulation of the PI3K/Akt/mTOR pathway is also involved in the inhibition of cell death by C1P, which can also contribute to cell survival by blocking the activity of the ceramide-generating enzymes acid sphingomyelinase (ASMase) and serine palmitoyl transferase (SPT). Moreover, C1P-promoted cell survival involves upregulation of inducible nitric oxide synthase (iNOS) and the subsequent production of nitric oxide (NO). Using photosensitive C1P analogues, it could be concluded that promotion of cell growth and inhibition of cell death were elicited by intracellularly generated C1P in a receptor-independent manner. The aim of the present review is to evaluate in detail the implication of the CerK/C1P axis in controlling cell proliferation and survival in mammalian cells, as well as to discuss and update on the molecular mechanisms by which C1P can accomplish these actions.

## 1. Introduction

Cell proliferation and survival are regulated by many hormones and growth factors (known as first messengers) of both protein and lipid nature, involving a number of signaling pathways. Most of these pathways are activated right after the interaction of first messengers with specific receptors that are usually located on the cell surface or in intracellular compartments, namely, the cytosol or the nucleus. Whilst hydrophilic agonists usually interact with cell surface receptors giving rise to the generation of second messengers, those that are hydrophobic mainly, but not exclusively, bind to intracellular receptors without the need to generate second messengers to elicit their regulatory actions. Instead, hydrophobic agonists, such as steroid hormones, vitamin D3, or retinoids, bind to specific response elements in the chromatin within the nucleus to regulate gene expression.

Besides these classical regulators of cell proliferation, some bioactive lipids, including the sphingophospholipid ceramide 1-phosphate (C1P), have also been shown to regulate cell growth and death [1,2]. C1P is mainly synthesized in the Golgi apparatus where ceramide kinase (CerK) activity is relatively high compared with that in other organelles or cell sites. CerK catalyzes the transfer of a phosphate group from ATP to ceramides that are synthesized de novo and transported by ceramide transfer protein (CERT) from the endoplasmic reticulum to the Golgi cisternae to form C1P (Figure 1). Major regulatory enzymes of the de novo synthesis pathway of ceramides include serine palmitoyl transferase (SPT) and ceramide synthases (CerSs). However, ceramides can also be synthesized by direct degradation of plasma membrane sphingomyelin (SM) by specific neutral or acidic sphingomyelinases (SMases) or from the recycling of sphingosine that is produced by degradation of complex sphingolipids in lysosomes, a pathway known as the salvage pathway (Figure 1). Once synthesized, C1P can be transported from the Golgi apparatus by a C1P transfer protein (CPTP) to different organelles, including the plasma membrane where it might participate in signal transduction events [3]. However, recent data from the Futerman lab indicate that, in contrast to their non-phosphorylated derivatives, the levels of C1P decreased in CerS2-null mice, with d18:1/C18:0-C1P decreasing by about 100-fold in the cerebellum, whereas the levels of C18-ceramide, and also C16-ceramide, were elevated in a CerS2-null mouse brain, suggesting that the pathway of C1P generation is regulated in such a way that the levels of C1P are not directly related to their precursor levels [4]. The purpose of the present review is to highlight and discuss the many roles of the CerK/C1P pathway in controlling cell proliferation and survival in mammalian cells as well as to review and update the molecular mechanisms by which C1P exerts these actions.

## 2. C1P as a Regulator of Cell Proliferation

Although CerK, the enzyme responsible for C1P biosynthesis in mammalian cells, was identified in brain tissue in 1989 [5], and C1P was further observed in human leukemia HL60 cells in 1990 [6], it was not until 1995 that C1P was first reported to be biologically active [7]. Specifically, synthetic short-chain N-acetyl-S1P (C2-C1P) and N-octanoyl-S1P (C8-C1P), which were enzymically synthesized using D-erythro-C2- or D-erythro-C8-ceramide as substrates, and diacylglycerol kinase (DAGK) were shown to potently stimulate DNA synthesis and cell division in rat-1 fibroblasts [7]. Also, C2-C1P and C8-C1P were chemically synthesized using D-erythro-ceramides without blocking of the 3-hydroxy group via monophosphitylation, and these synthetic C1P species were equally potent at stimulating cell proliferation [8]. Further studies using natural (long-chain) C1P showed pronounced stimulation of cell growth in EGFR T17 cells [9], which are NIH 3T3 fibroblasts overexpressing the EGF receptor, thereby confirming the mitogenic effects of the short-chain C1Ps in fibroblasts. Interestingly, either short- or long-chain ceramides, which are potent inducers of apoptosis [10], were able to completely block C1P-stimulated DNA synthesis and cell division [7,9], pointing to a critical role of CerK and C1P phosphatases in the regulation of cell homeostasis.

Until recently, phosphorylation of ceramide by CerK was the only mechanism by which C1P was produced in mammalian cells. However, the fact that CerK-knockout mice showed high levels of C1P led to the discovery of an alternative pathway for C1P biosynthesis. This pathway involved the intervention of diacylglycerol kinase-ζ (DAGK-ζ) [11], an enzyme that can phosphorylate both diacylglycerol (DAG) and ceramides in vitro [6]. Nonetheless, the low ability of DAGK-ζ to phosphorylate ceramides results in only a small amount of additional C1P being synthesized, which may not be sufficient to account for the relatively high amount of C1P that is still present in the CerK (-/-) cells. An alternative mechanism for producing C1P might be the SMase D pathway, as is the case for some arthropods and bacteria [12,13]. SMase D is a phospholipase D (PLD)-type phospholipase that breaks down SM to C1P and choline. However, no SMase D activity has so far been identified in mammalian cells. Another potential pathway for C1P biosynthesis might involve acylation of S1P by an acyl transferase, but no enzyme capable of catalyzing the transfer of an acyl fatty acid chain to the S1P moiety has ever been reported.

Investigation into the mechanisms by which C1P stimulates cell proliferation revealed that treatment of primary, bone marrow-derived, macrophages (BMDMs) with C1P rapidly stimulated phosphorylation of the mitogen-activated protein kinases (MAPKs) ERK1-2 and JNK, as well as the downstream phosphatidylinositol 3-kinase (PI3K) effector Akt (also known as protein kinase B, PKB). The implication of these pathways in the mitogenic effect of C1P was studied using pharmacological inhibitors of these kinases, as well as specific siRNA to silence the genes encoding each particular kinase [14]. C1P-stimulated cell proliferation was also regulated by glycogen synthase kinase 3β (GSK3β), the product of retinoblastoma gene, c-Myc, and nuclear factor kappa B (NF-κB), a transcription factor that is also involved in cell survival and inflammation [15], as discussed below.

A relevant downstream target of ERK1-2 and Akt is the mammalian (mechanistic) target of rapamycin (mTOR), which is an important regulator of protein synthesis, autophagy, and cell proliferation and is also involved in CD8 T-cell differentiation [16]. The mTOR kinase exists in the form of two different protein complexes, mTORC1 and mTORC2. The mTORC1 complex is composed of mLST8 (mammalian lethal with SEC13, PRAS40 (proline-rich Akt substrate of 40 kDa)), Raptor (regulatory-associated protein of mTOR), and active (GTP-bound) Rheb (Ras homologue enriched in brain) G protein, and it usually signals through phosphorylation of p70S6K. In turn, the latter kinase phosphorylates ribosomal protein S6, an important regulatory element of translation, and regulates eukaryotic translation initiation factor (eIF-4E)-binding protein [17,18]. The mTORC2 complex is a rapamycin-insensitive serine/threonine kinase that also participates in the regulation of cell proliferation and survival and is triggered by stress stimuli [19]. The mTORC2 components are rictor (rapamycin-insensitive companion of mammalian target of rapamycin), SIN1 (stress-activated protein kinase-interacting protein 1), and mLST8, and its substrates include Akt, protein kinase C (PKC), and serum/glucocorticoid-regulated kinase (SGK), which are members of the group of kinases related to PKA, PKG, and PKC (AGC) protein kinase family. In particular, C1P caused phosphorylation of the mTORC1 component PRAS40 to activate the mTOR pathway leading to stimulation of cell proliferation [20]. C1P was also able to stimulate another Ras homologue gene family member termed RhoA. In particular, treatment of macrophages with C1P caused rapid phosphorylation of RhoA, leading to activation of ROCK (Rho-activated kinase). Inhibition of ROCK caused a potent reduction in C1P-stimulated DNA synthesis and cell division, suggesting that RhoA/ROCK is also an essential pathway in this process [20]. The latter observations are in agreement with recent findings showing that RhoA promotes the proliferation of melanoma cells [21], an action involving stimulation of glucose uptake, which is consistent with the stimulation of glucose uptake and metabolism by C1P in macrophages [22]. Moreover, RhoA promoted epidermal stem cell proliferation [23], and overexpression of RhoA stimulated proliferation of cervical cancer cells [24].

Another important pathway that was implicated in the stimulation of macrophage proliferation by C1P was the DAG/PKC pathway. In cell signaling processes, DAG is usually generated by the action of phosphatidylinositol-dependent phospholipase C (PI-PLC) or by phosphatidylcholine (PC)-dependent PLC. However, no inositol trisphosphate (IP3) or phosphocholine (P-Chol) was generated by the action of C1P in macrophages, suggesting that none of these phospholipases were involved in the mitogenic effect of C1P [25]. However, C1P was able to stimulate sphingomyelin synthase activity (SMS), which catalyzes the transfer of P-Chol from PC to ceramide producing the physiologic PKC activator DAG. Of relevance, inhibition of C1P-stimulated SMS activity resulted in the complete inhibition of C1P-stimulated cell division, suggesting that SMS is the enzyme responsible for PKC activation by C1P in macrophages. The PKC isoform that was stimulated by C1P was PKC-α [25]. This PKC isoform was also involved in the production of reactive oxygen species (ROS) in macrophages. Specifically, low levels of ROS were able to mediate the mitogenic effect of C1P in these cells [26]. ROS generation involved prior activation of multi-subunit NADPH oxidase (NOX), which was preceded by activation of PKC-α and cytosolic calcium-dependent phospholipase A2 (cPLA2). In fact, PKC-α or cPLA2 inhibitors abrogated the stimulation of cell proliferation by C1P, further emphasizing the importance of these enzymes in C1P-regulated mitogenesis [26]. Further investigation into the mechanisms by which C1P promoted cell growth revealed that both vascular endothelial growth factor (VEGF) [27] and LPA [28] mediated the stimulation of cell proliferation by C1P. Specifically, treatment of macrophages with C1P caused the release of VEGF into the extracellular milieu allowing the growth factor to interact with its VEGF receptor-2 isoform in an autocrine/paracrine manner to stimulate cell proliferation [27]. C1P-stimulated VEGF release was dependent upon stimulation of the PI3K/Akt-1 and ERK1-2 pathways, as inhibition of these kinases with selective pharmacological inhibitors or with specific gene-silencing siRNA abrogated VEGF release [27]. In myoblasts, however, C1P stimulated cell proliferation via the LPA signaling axis involving LPA receptors 1 and 3 [28]. Nonetheless, the mitogenic effects of C1P were not restrictive to fibroblasts, myoblasts, or macrophages. In fact, various research groups have confirmed the regulation of cell proliferation by C1P in other cell types. For example, work from the Rotstein lab demonstrated that C1P potently induces DNA synthesis and proliferation of retina photoreceptors, pointing to a relevant role of C1P in vision physiology [29], and Mitra and co-workers and Pastukhov and co-workers showed that CerK played a relevant role in regulating breast and lung cancer cell proliferation [30,31]. Regarding breast cancer, CerK was shown to be upregulated in metastatic breast cancer cells [32] and is a common therapeutic target for both triple-positive and triple-negative breast cancer [33]. CerK was also associated with a poor prognosis in breast cancer patients [34], and endocrine therapy-resistant breast cancer cells were more sensitive to CerK inhibition than therapy-sensitive breast cancer cells [35]. Also, CerK contributed to proliferation of renal mesangial cells and fibroblasts [36]. Moreover, CerK inhibition suppressed growth and induced apoptosis in cisplatin-resistant ovarian cancer cells, thereby enhancing the efficacy of cisplatin as chemotherapeutic agent in this type of cancer [36].

Another type of cancer in which CerK/C1P plays a critical role is prostate cancer [37]. In this connection, it has been recently shown that the PPARβ/CerK/C1P signaling pathway is responsible for stimulation of prostate cancer cell growth by exposure to antimony, an industrial heavy metal pollutant [38]. C1P also enhanced cell growth in Kaposi sarcoma cells [39]. The mitogenic action of C1P was also demonstrated in human neuroblastoma cells treated with the active form of vitamin D3 (1,25(OH)2D3) or with synthetic analogues of this vitamin/hormone with low ability to alter calcium metabolism, such as ZK191784. The antiproliferative effects of these agonists were associated with CerK inhibition and depletion of C1P levels in these cells [40].

Besides CerK, CPTP also plays a relevant role in the regulation of cell growth. In fact, it was recently reported that CPTP is highly expressed in pancreatic cancer and was associated with poor prognosis in patients with this type of cancer [41]. Also, CerK is highly expressed in pancreatic cancer cells, and inhibition of this kinase substantially decreased pancreatic cancer cell migration [42]. Although C1P can be found in extracellular compartments, using photosensitive caged C1P analogues, which are cell permeable compounds that bypass cell plasma membrane receptors and can be released intracellularly upon visible light irradiation, it was demonstrated that stimulation of cell proliferation was caused by endogenous C1P, independently of extracellular C1P [8,43]. Nonetheless, extracellular C1P is also relevant in cell biology, as it can regulate glucose uptake and cell migration in a receptor-mediated manner, as discussed below.

## 3. C1P as a Regulator of Cell Death and Survival

Further investigation into the processes by which C1P regulates cell homeostasis in mammalian cells revealed a new biological activity of C1P, this being the inhibition of cell death and the promotion of cell survival. Initial studies were carried out using primary BMDMs, as these cells are absolutely dependent on macrophage colony-stimulating factor (MSCF) for viability and growth [44]. Incubation of BMDMs in the absence of MCSF caused a significant decrease in cell viability, which was accompanied by a substantial increase in intracellular ceramide concentration [45] and a sharp decrease in the endogenous levels of C1P followed by upregulation of cytochrome c release, phosphatidylserine externalization, activation of caspases 9 and 3, and DNA fragmentation, all being hallmarks of apoptosis [46]. Noteworthily, inclusion of C1P in the MCSF-free culture medium reestablished the intracellular levels of C1P and prevented the macrophages from entering apoptosis. Specifically, C1P blocked ceramide accumulation, activation of the caspase 9–caspase 3 cascade, and DNA fragmentation in the primary BMDMs [46]. Of interest, ceramide accumulation in the apoptotic macrophages was due to potent upregulation of acidic SMase (ASMase), a major ceramide-producing enzyme in these cells [45]. The mechanism by which C1P decreased ceramide levels involved potent blockade of ASMase, which probably occurred by physical interaction of C1P with the enzyme. The latter action could release ASMase from inhibition, leading to ceramide generation and apoptotic cell death [46]. An analogous mechanism was observed in rat alveolar macrophages (NR8383 cells), which do not depend on MCSF for growth or viability. Ceramide generation in these cells incubated under apoptotic conditions (absence of serum in the culture medium) was mainly due to upregulation of SPT [47], a key regulatory enzyme of de novo synthesis of ceramides [48,49,50]. Importantly, C1P completely blocked SPT activity bringing ceramides down to normal levels and promoting cell survival [47]. The antiapoptotic effect of C1P in these cells was accompanied by blockade of cytochrome c release and inhibition of Bax, which is a proapoptotic protein of the Bcl2 family that plays a critical role in the mitochondrial intrinsic pathway of apoptosis [47]. The antiapoptotic effects of C1P also involved stimulation of the PI3K/Akt pathway, which is a major mechanism by which growth factors promote cell survival [51]. Also, C1P upregulated the activity of NF-κB, a transcription factor that is downstream of PI3K/Akt in the cascade of events leading to cell survival, and was also able to counteract the depletion of antiapoptotic Bcl-xl, a protein that is heavily depleted under apoptotic conditions [51]. A downstream target of PI3K/Akt is the inducible form of nitric oxide synthase (iNOS), an enzyme responsible for the production of nitric oxide (NO) in many cell types. Interestingly, C1P promoted NO synthesis leading to blockade of apoptosis, and inhibition of iNOS prevented C1P from promoting cell survival [52]. The involvement of CerK/C1P in the promotion of cell survival was also demonstrated by Mitra and co-workers using human A549 lung cancer cells [30] and by Payne and co-workers who showed that CerK promoted breast cancer cell survival and mammary tumor recurrence [53]. Concerning cancer cells, engineered nanomicelles targeting proliferation and angiogenesis inhibited tumor progression in colorectal cancer models by impairing the synthesis of C1P [54]. Also, in addition to stimulating proliferation of retina photoreceptors, C1P promoted their survival and prevented their degeneration [29]. C1P also protected endothelial colony-forming cells from apoptosis and increased vasculogenesis in vitro and in vivo [55], and CerK was implicated in protection of astrocytes and neurons against ceramide-induced cell death [56].

Concerning neuroprotection, C1P increased p-glycoprotein transport activity at the blood–brain barrier via prostaglandin E2 signaling processes [57]. Also, C1P was shown to increase after posttraumatic brain injury [58], an action that may be associated with inflammatory responses that are triggered in injured tissue. Nonetheless, although C1P has proinflammatory properties [59], it can protect against inflammation in some cell types or tissues including peripheral blood mononuclear cells [60] or in lung tissue. In particular, C1P plays a relevant role in protection against pulmonary emphysema, a disease associated with chronic obstructive pulmonary disease and destruction of lung parenchyma and in tobacco smokers [61] and blocked the production of proinflammatory IL-8 in human neutrophils that were previously challenged with lipopolysaccharide (LPS) [62]. Also, C1P substantially decreased neutrophil infiltration into the lungs of LPS-treated mice [62]. Also, importantly, C1P was shown to protect against cyclophosphamide-induced ovarian damage in a mice model of premature ovarian failure [63]. Cyclophosphamide is a drug used for the treatment of various types of cancer, which may cause sterility in men and women. In particular, treatment with cyclophosphamide increased the expression of proapoptotic Bax and caused a substantial decrease in antiapoptotic Bcl-xl protein leading to ovarian damage. Administration of C1P restored the levels of both proteins, thereby protecting against ovarian damage [63]. Also, cyclophosphamide increased the expression of ASMase leading to accumulation of proapoptotic ceramide in the ovaries, and treatment with C1P decreased ASMase expression, promoting cell survival in a similar way as in macrophages [46,63]. More recently, it has been suggested that CerK plays a protective role against oxidative damage in the liver [64]. Strong support for the later observations has been recently provided by Tao et al., who have demonstrated that CerK suppresses ferroptosis and protects against alcohol-associated liver disease, an action that involved prior activation of the p38 MAPK/heat shock protein beta-1 (HSPB1) pathway [65], thereby reinforcing the notion that C1P is a crucial regulator of cell survival.

## 4. Other Major Biological Actions of C1P

In addition to promoting cell growth and survival, C1P is implicated in the regulation of other relevant cell functions, at both physiologic and pathologic settings. A major role of the CerK/C1P axis in cell biology is its participation in inflammatory responses. As mentioned above, C1P promotes inflammation in different cell types [59] and is involved in neutrophil degranulation [66] and bacterial proliferation [67]. However, C1P also has anti-inflammatory properties, especially in the lungs [61,62], and potently inhibited lipopolysaccharide (LPS)-induced production of interleukin (IL)-6, IL-8, and IL-1b in human peripheral blood mononuclear cells [60].

Another important role of CerK/C1P is the regulation of cell migration. Initial studies using different human or mouse macrophage cell lines demonstrated that C1P has chemotactic properties, being able to potently stimulate cell migration [68,69]. Of note, C1P-stimulated cell migration required prior interaction of C1P with a Gi protein-coupled receptor (Figure 1) that was partially characterized in macrophages [68,69]. Subsequent studies showed that both CerK and extracellular C1P stimulated human pancreatic cancer cell migration and invasion, actions that involved activation of various cell signaling pathways, including MEK/ERK1-2, PI3K/Akt/mTOR, MCP-1 release, and upregulation of metalloproteinases 2 and 9 [68,69,70]. The chemotactic properties of C1P have been confirmed by different investigators using normal as well as transformed (carcinogenic) cells [71,72,73,74,75,76,77,78,79,80]. Nonetheless, in A549 human lung adenocarcinoma cells, CerK negatively regulates cell migration [81] a finding that is in agreement with the anti-inflammatory actions of C1P observed in lung tissue [82,83].

CerK and intracellularly generated C1P have also been involved in promoting the differentiation of preadipocytes into mature adipocytes [84], whereas extracellular C1P inhibited this process [85]. The latter action occurred in a receptor-dependent manner, pointing to a relevant role of exogenous C1P in maintaining an appropriate balance of adipocyte maturation.

C1P has also been shown to regulate intracellular calcium concentration in Jurkat T cells [86], and ceramide and C1P regulate sperm motility and acrosomal exocytosis through modulation of calcium levels, which is relevant for oocyte fertilization. In particular, C1P facilitates calcium influx through various channels, enabling the acrosome reaction in response to stimuli like progesterone [87]. Also, C1P played a protective role in oocyte health and fertility, particularly in the context of chemotherapy-induced ovarian damage [63].

## 5. Concluding Remarks

The intracellular generation of C1P is a key factor for regulation of cell growth and death. While the mitogenic actions of C1P involve activation of a variety of kinases that are known to regulate cell proliferation, C1P-promoted cell survival involves inhibition of two major ceramide-generating enzymes, namely, ASMase and SPT, as well as blockade of many proapoptotic proteins, upregulation of NO production, and activation of the PI3K/Akt pathway. Nonetheless, not all of the cellular C1P species may work in the same direction or exert similar actions. In fact, the short-chain acetyl-C1P species is much less effective at inhibiting ASMase than the long-chain C1P species [46]. Also, it is not clear whether the degree of unsaturation of the N-bound fatty acids in the ceramide moiety of C1P would cause any changes in the bioactivity of the different C1P species. In addition, the effects of C1P are cell or tissue specific, adding complexity to understanding the role played by CerK/C1P in cell biology. The use of sophisticated techniques, such as high-resolution accurate-mass spectrometry with high mass resolving power in conjunction with specific animal models would be key to understanding the role played by C1P in physiology and disease. All of these important aspects of C1P biology await further investigation to be elucidated.

## Figures and Tables

**Figure 1 ijms-26-08374-f001:**
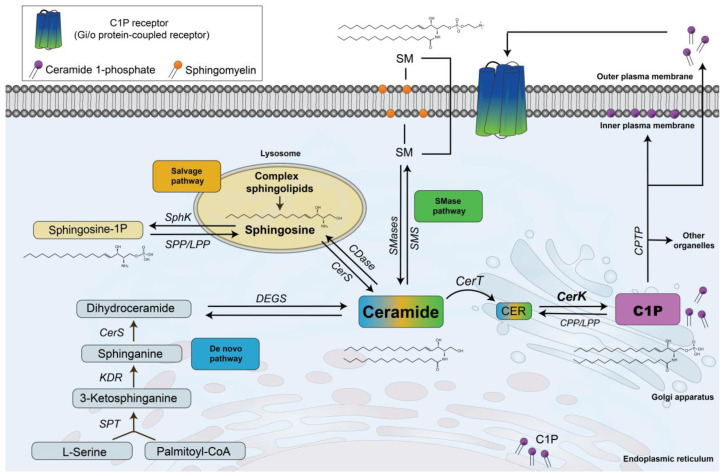
Biosynthesis of ceramide 1-phosphate (C1P) in mammalian cells. C1P is mainly synthesized in the Golgi apparatus from ceramide that is transported by ceramide transfer protein (CerT) from the endoplasmic reticulum, where ceramides are synthesized de novo. Alternatively, ceramides can be generated by the SMase pathway from direct degradation of sphingomyelin (SM) or by the salvage pathway from the metabolism of complex sphingolipids. C1P is also present in the perinuclear membrane. Once synthesized, C1P can be transported by ceramide 1-phosphate transfer protein (CPTP) to the plasma membrane and other organelles. Also, C1P can be secreted to the extracellular milieu and bind to a putative Gi/o protein-coupled receptor located in the plasma membrane of cells. CDase, ceramidase; CerS, ceramide synthase; CER, ceramide; CerT, ceramide transfer protein; CPP, C1P phosphatase; DEGS, dihydroceramide desaturase; KDR, ketosphinganine reductase; LPP, lipid phosphate phosphatase; SMase, sphingomyelinase; SMS, sphingomyelin synthase; SphK, sphingosine kinase; SPP, S1P phosphatase; SPT, serine palmitoyl transferase.

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
