# Peer review of "Role of Ceramide Kinase/C1P in the Regulation of Cell Growth and Survival"

_ijms, 2025, doi:10.3390/ijms26178374_

Round 1
Reviewer 1 Report
Comments and Suggestions for Authors
In this review the authors discuss the role of ceramide kinase and C1P in cell survival and proliferation.
- I feel that the aim of the review should also be mentioned in the abstract.
- What I really missed is a reference to the novelty of the study. The authors should refer to the main focus of the current review that differentiates it from recent reviews of the same group (e.g. Cell Signal. 2021 Jul:83:109980.doi: 10.1016/j.cellsig.2021.109980) or of the Chalfant’ group (Adv Exp Med Biol. 2019:1159:65-77.doi: 10.1007/978-3-030-21162-2_5). Does the current review mainly focus on normal cells? If this is the case, references to cancer cells could be omitted or be included as extra paradigms. Or does the current review present -along with classical molecular pathways- alternative, non-canonical pathways through which ceramides act to exert their effects on cell survival and growth? This could be explicitly stated and emphasized by, for example, altering the review’s structure and devoting to this a separate section.
- Bibliography is quite extensive for the length of the review. It could be shortened by being restricted (wherever possible) to the more recent literature.
- Please check the formatting of the References section: Some studies appear with incomplete information (e.g. references 26, 32, 39, 41, etc).
Minor points:
- Line 61: Delete “and”.
- Line 94: Please add a comma after “synthesized”.
Author Response
RESPONSES TO THE COMMENTS OF REVIEWER 1
REVIEWER 1:
In this review the authors discuss the role of ceramide kinase and C1P in cell survival and proliferation.
1. I feel that the aim of the review should also be mentioned in the abstract.
RESPONSE: We agree with the reviewer. Accordingly, a new statement on the aim of the review has been added to the abstract. Please see lines 31-34.
2. What I really missed is a reference to the novelty of the study. The authors should refer to the main focus of the current review that differentiates it from recent reviews of the same group (e.g. Cell Signal. 2021 Jul:83:109980.doi: 10.1016/j.cellsig.2021.109980) or of the Chalfant’ group (Adv Exp Med Biol. 2019:1159:65-77.doi: 10.1007/978-3-030-21162-2_5). Does the current review mainly focus on normal cells? If this is the case, references to cancer cells could be omitted or be included as extra paradigms. Or does the current review present -along with classical molecular pathways- alternative, non-canonical pathways through which ceramides act to exert their effects on cell survival and growth? This could be explicitly stated and emphasized by, for example, altering the review’s structure and devoting to this a separate section.
RESPONSE: Our 2021 review published in “Cellular Signalling” that the reviewer refers to is mainly focused on the implications of C1P in lung cancer cell growth and dissemination. In the present review we have addressed the regulation of cell growth and survival by C1P in a broader way and have discussed more deeply on the mechanisms that are implicated in these processes. Also very importantly, we have thoroughly updated the information on these topics by including the most recent publications that are available in the scientific literature.
The excellent review published by Chalfant and co-workers in 2019 is obviously out of date, after 6 years since it was published. In the present review, we have updated in detail on the regulation of cell growth and survival by CerK/C1P, as mentioned above.
We have not differentiated on the control of cell proliferation and survival by CerK/C1P between normal and cancer cells because most of the major signaling pathways that are known to be regulated by CerK/C1P are the same in both cell growth models. We believe that by not discriminating between the two models, we can have a broader view of the mechanisms that are triggered or blocked by C1P to control cell growth and survival.
3. Bibliography is quite extensive for the length of the review. It could be shortened by being restricted (wherever possible) to the more recent literature.
RESPONSE: We agree with the reviewer on that references corresponding to a particular subject should be as recent as possible. However, we also believe that credit should preferentially be given to authors who first reported on a particular aspect of a subject, and also to the scientists who contributed importantly to that particular subject. Nonetheless, to meet the reviewer´s request we have thoroughly examined the manuscript and have eliminated nine references, which we believe are not essential to the manuscript. The total number of references is now 86 instead of 95.
4. Please check the formatting of the References section: Some studies appear with incomplete information (e.g. references 26, 32, 39, 41, etc).
RESPONSE: We have used the MDPI format in the Endnote application to list the references in the manuscript, but are unable to detect any lack of information in the mentioned references. Specifically, ref 26 is published as a pdf of the accepted manuscript prior to publication. There is no additional information on this particular reference. Concerning refs 32, 39, 41, or others, we were also unable to identify any missing information, so we kindly ask the reviewer to please be more concise and let us know of the sort of information that he/she sees missing. Thank you so much.
Minor points:
- Line 61: Delete “and”.
- Line 94: Please add a comma after “synthesized”.
RESPONSE: Both minor points have been corrected.
Reviewer 2 Report
Comments and Suggestions for Authors
Dear Authors,
Ceramide is an important bioeffector molecule. This review titled: Role of ceramide kinase/C1P in the regulation of cell growth 2
and survival is a very well written topic.
The figure 1 depicting the ceramide-1-phosphate in mammalian cells is very well thought out and drawn figure. It is the main message of the whole review.
The preclinical, invitro studies of C1P is discussed very in-dept. C1P is relevance to several diseases including cancer. The identified refs., provide very indepth data. I felt, if the biosynthesis would have been with structures, it would capture med.chem attention.
Ceramide biosynthesis and metabolism is very relevant. There are several hits identified in these enzymes. These were also not listed. It covers deep signalling pathways. I enjoyed reading.
Author Response
RESPONSES TO THE COMMENTS OF REVIEWER 2
REVIEWER 2
Dear Authors,
Ceramide is an important bioeffector molecule. This review titled: Role of ceramide kinase/C1P in the regulation of cell growth and survival is a very well written topic.
The figure 1 depicting the ceramide-1-phosphate in mammalian cells is very well thought out and drawn figure. It is the main message of the whole review.
The preclinical, invitro studies of C1P is discussed very in-dept. C1P is relevance to several diseases including cancer. The identified refs., provide very indepth data. I felt, if the biosynthesis would have been with structures, it would capture med.chem attention.
Ceramide biosynthesis and metabolism is very relevant. There are several hits identified in these enzymes. These were also not listed. It covers deep signalling pathways. I enjoyed reading.
RESPONSE: We thank the reviewer for the positive comment on the written topic of the review, and on Figure 1.
We also thank the reviewer for the positive comments on the discussion on pre-clinical and in vitro studies, and for acknowledging that the references that have been selected provide in-depth data. With regards to including the structures of the molecules, following the reviewer´s indications, in Figure 1 we have now included the chemical structures of the relevant metabolites that are synthesized in the different pathways and are the focus of the present review. These are the four bioactive sphingolipids: sphingosine, S1P, ceramide and C1P, as well as SM, which is a precursor of ceramide and is the most abundant sphingolipid in mammalian cells. We have tried not to overload the figure with the structures of metabolic intermediates that are not relevant to the focus of the review, so as to keep the figure as clear as possible.
We particularly thank the reviewer for bringing our attention on the enzymes that were not listed, so in the legend to Figure 1 we have now included the enzymes that were missing in the original version of the manuscript (Sphingomyelinase, SMase, and sphingomyelin synthase, SMS). We have also noticed some typographical errors, which have now been corrected. Please see lines 88-90.
We thank the reviewer very much for the helpful and constructive comments, and for the general positive feed-back on our manuscript.
Round 2
Reviewer 1 Report
Comments and Suggestions for Authors
This is the revised version of a previously submitted manuscript. The authors have addressed some of my concerns or provided convincing aetiology on why they did not follow some of my suggestions. Regarding my comment on references' formatting, in the previous version only doi appeared for some articles, but now this problem seems to have been resolved.